# OpenReview forum: "Who Measures What Matters? An Analysis of Social Impact Evaluations in Foundation Model Reporting"
_ICLR.cc/2026/Conference — ICLR 2026 Conference Desk Rejected Submission_

### Official Review · Reviewer_q1un · 2025-10-28

**Soundness:** 3
**Presentation:** 3
**Contribution:** 3
**Rating:** 4
**Confidence:** 4

**Summary:**

The paper offers a large‐scale, cross‑provider analysis of how social impact evaluations for foundation models are reported. The authors examine seven base‑level dimensions, bias/representational harms, sensitive content, performance disparity, environmental costs, privacy/data protection, financial costs, and data/moderation labo, drawing on first‑party release‑time reports and post‑rlease third‑party sources. They complement the quantitative analysis with semi‑structured interviews of developers to explain observed gaps and incentives.

Contributions:
(1) The first broad, quantitative snapshot of social‑impact evaluation reporting across providers and categories; (2) qualitative context via developer interviews; (3) commitment to release an annotated dataset post‑publication (Sec. 1, bullets; pp. 1–2).

**Strengths:**

1- Timely and policy‑relevant question with clear framing around seven base‑level social‑impact dimensions (Sec. 4. This focuses the analysis on what model developers control pre‑deployment.

2 - Cross‑provider, multi‑region, multi‑sector dataset; per‑category scoring distinguishes mere mentions from methodologically detailed reporting (Fig. 1–6).

3 - Methodological care. Bayesian hierarchical ordinal regression is an appropriate choice for 0–3 scales; reporting of priors, cutpoint parameterization, and diagnostics inspires confidence (App. 8.7; Table 2).

4 - Strong empirical evidence that third‑party evaluations carry much of the social‑impact load and that environmental and bias reporting has decreased since 2023Q3 (Fig. 2; Fig. 14d).

5 - Mixed‑methods triangulation. Interviews provide plausible explanations for gaps (organizational incentives, reputational/legal risk, lack of usable frameworks) that align with the quantitative patterns.

**Weaknesses:**

Major:

- Annotation reliability not quantified. The paper explicitly states no formal inter‑annotator agreement was computed (App. 8.3, p. 21)
- Counting inconsistency in source totals. The abstract reports 99 first‑party and 187 post‑release sources; Sec. 1 lists 132 first‑party and 174 third‑party/leaderboards (pp. 1–2). This raises questions about inclusion criteria, deduplication, and whether leaderboards are counted as “post‑release sources.” Actionable: reconcile counts, define each bucket precisely (tech report, model card, system card, blog, leaderboard, peer‑reviewed paper), and provide a consolidated PRISMA‑style flow diagram.

- Third‑party papers are restricted to peer‑reviewed works from 2024 onward (Sec. 3), potentially under‑representing earlier or non‑archival evaluations, and privileging Anglophone venues. This matters for temporal claims (spikes/declines in 2024–2025) and for regional equity.

**Questions:**

1 Reliability. Did you pilot the rubric on a shared subset and, if so, what was the observed IAA? If not, could you compute α/κ on 10–20% of items and report robustness of key conclusions when restricting to items with coder agreement?

2 Temporal analyses. How sensitive are the “decline since 2023Q3” findings (Fig. 2; Fig. 14) to (a) collapsing variants to base models, and (b) including non‑peer‑reviewed third‑party sources pre‑2024?

3 Provider weighting. Beyond hierarchical intercepts, did you try provider‑level weighting (e.g., one vote per provider‑release) to mitigate large‑family effects (Gemini/Llama)? If tried, did conclusions change? (Sec. 3)

---

> ### Author Response · Authors · 2025-12-03
> **Thank you!**
>
> We thank the reviewer for the positive feedback, in particular that they recognize our work as “timely and policy-relevant” and that they highlight our “methodological care” alongside the “mixed-methods triangulation” and the overall “strong empirical evidence” for our findings.
>
> **1.  Missing annotation reliability**
>
> We thank the reviewer for pointing out that we missed stating the annotation reliability. We have now added inter-annotator agreement for 10% of the items, each independently double-annotated by two annotators. Krippendorff’s alpha with ordinal weighting was 0.752 for first-party reports and 0.831 for third-party reports.
>
> **2. Inconsistent counts**
>
> We also thank the reviewer for the constructive feedback regarding the stated counts. In particular, we agree that our wording was ambiguous – as suggested, we have reconciled counts (both are higher now because, in response to your third point, we have added annotations for sources from pre-2024) and we have added a flow diagram to the appendix to visualize the inclusion process.
>
> **3. Restriction of third-party sources to pre-2024**
>
> We appreciate the reviewers’ feedback regarding the inclusion of evaluations pre-2024. We have expanded the dataset and annotations to include evaluations of first- and third-party reporting on foundation models between 2018 and 2023, resulting in 64 additional annotated sources, and we have found that the temporal effect has become more pronounced.
>
> **4. Questions regarding the sensitivity of results**
>
> In our hierarchical model, group-level intercepts are included for region, county and provider arranged as nested deviation (i.e. provider within country, country within region), which means that providers with many releases don’t just drag the global average but instead get their own random effect that gets partially pooled toward the overall distribution. Under this setting, the negative effect of first-party vs. third-party (-2.22 to -4.67 LOR) is large compared to typical hierarchical shrinkage effects, which cannot be reversed except by very extreme reweighting. We experimented with different hierarchical prior SDs (0.4 vs 0.25 using the non-centred parametrization, the former implying $\pm0.8$ variation on the LOR scale, $\pm0.5$ for the latter), as well as adding group-level slopes at the level of providers, and the results were similar.
>
> We also want to reiterate our appreciation for your detailed comments and suggestions – they helped us to significantly improve our work! Thank you!

---

### Official Review · Reviewer_yzxU · 2025-11-01

**Soundness:** 3
**Presentation:** 1
**Contribution:** 3
**Rating:** 6
**Confidence:** 4

**Summary:**

This is a nice paper that presents an annotated database of model evaluations in the context of evaluation for social impact. The database is a valuable contribution, but I believe the analysis could be improved, and the figures should be improved.

**Strengths:**

- Important topic.
- The annotated dataset is a valuable contribution.

**Weaknesses:**

Figures: I found the figures to be very suboptimal for conveying the desired information. For example, red is usually used to highlight something undesirable, but in some figures, it shows the best score for evaluations. In some of the figures, the test is too small and isn't clear even when you zoom in (and will be unreadable if printing). I think the figures / tables showing progress over time will be much more effective if presented as a figure with 't' on the x-axis.

Summary statistics: I really think the main body of the paper will benifit from summary statistics of the dataset.

 Analysis: the two main conclusions 1) the third-party analysis tends to focus better on social impact, and 2) certain factors, such as bias, are more investigated in the climate impact and moderation -- both somewhat unsurprising. This does not mean that there is no value in verifying them, but I am certain the data contains more novel insights as well, which could be highlighted.

Policy: the paper does not say anything about policy -- which is very related to the quality of 1st party evaluations.

**Questions:**

Please state if and how you think you can improve the paper with respect to the weaknesses highlighted.

---

> ### Author Response · Authors · 2025-12-03
> **Thank you!**
>
> We sincerely thank the reviewer for their careful and constructive assessment of our work. We greatly appreciate the reviewer’s recognition that the paper addresses an “important topic” and that “the annotated dataset is a valuable contribution.” We also thank the reviewer for acknowledging that our dataset and analysis meaningfully advance understanding of model evaluation practices in the context of social impact assessment. We are grateful for the reviewer’s time, insight, and detailed suggestions, which have strengthened the revised manuscript.
>
> **1. Figures**
>
> We thank the reviewer for this helpful feedback on figure design and clarity. We agree that improving the visual expressiveness and readability of the figures will substantially strengthen the paper.
> In the updated version (see revised pdf), we have replaced red with green so that high-quality evaluation scores are not encoded using a color typically associated with error or risk. We have also increased font sizes to ensure better readability, as suggested by the reviewer. We thank the reviewer again for their constructive suggestions; they resulted in clearer and more intuitive visualizations.
>
> **2. Summary statistics**
>
> We thank the reviewer for highlighting that summary statistics would strengthen the main text. In the revision, we have added a summary-statistics section which includes the overall distribution of scores across all 7 dimensions (mean, median, variance), the proportion of first-party vs. third-party reports by evaluation type, the top-level missingness rates per dimension (e.g., privacy reported in X% of first-party reports, environmental costs in Y%), and the average number of social-impact dimensions evaluated per report, broken down by sector and region (e.g., academia vs. industry). Please also find these statistics below.
>
>
> _Overall distribution_
>
> | **category** | **mean** | **median** | **var** |
> | ------------ | -------- | ---------- | ------- |
> | 1            | 2.246    | 3.0        | 1.380   |
> | 2            | 2.446    | 3.0        | 0.771   |
> | 3            | 2.239    | 3.0        | 1.447   |
> | 4            | 1.584    | 2.0        | 1.481   |
> | 5            | 1.351    | 1.0        | 1.800   |
> | 6            | 0.779    | 1.0        | 0.354   |
> | 7            | 0.124    | 0.0        | 0.228   |
>
> _Proportion of first-party vs. third-party reports by category_
>
> | **category** | **is_first_party** | **is_third_party** | **is_first_party_proportion** |
> | ------------ | ------------------ | ------------------ | ----------------------------- |
> | 1            | 219                | 465                | 0.320                         |
> | 2            | 282                | 1388               | 0.169                         |
> | 3            | 224                | 386                | 0.367                         |
> | 4            | 214                | 154                | 0.582                         |
> | 5            | 208                | 162                | 0.562                         |
> | 6            | 189                | 164                | 0.535                         |
> | 7            | 186                | 0                  | 1.000                         |
>
> _Top-level missingness rate per dimension_
>
> | **category** | **n_missing** | **n_total** | **proportion_missing** |
> | ------------ | ------------- | ----------- | ---------------------- |
> | 1            | 125           | 186         | 0.672                  |
> | 2            | 93            | 186         | 0.500                  |
> | 3            | 121           | 186         | 0.651                  |
> | 4            | 109           | 186         | 0.586                  |
> | 5            | 154           | 186         | 0.828                  |
> | 6            | 100           | 186         | 0.538                  |
> | 7            | 171           | 186         | 0.919                  |
>
> _Average number of social-impact dimensions evaluated per report, broken down by sector and region_
>
> | **sector** | **average** |
> | ---------- | ----------- |
> | Academia   | 1.573       |
> | Government | 1.538       |
> | Industry   | 1.463       |
> | Nonprofit  | 1.400       |
>
>
> | **region**                   | **average** |
> | ---------------------------- | ----------- |
> | East Asia                    | 1.361       |
> | Europe                       | 1.445       |
> | Middle East and North Africa | 1.593       |
> | North America                | 1.493       |
> | South and Central Asia       | 2.333       |

---

> > ### Author Response · Authors · 2025-12-03
> >
> > **3. Analysis: Reviewer finds the two main conclusions unsurprising**
> >
> > While some readers may intuitively expect that third-party evaluations are more thorough, or that bias is examined more frequently than climate or labor impacts, these assumptions have, until now, rested almost entirely on anecdotal or case-specific impressions. A core contribution of our paper is to replace these long-standing assumptions with systematic, quantitative evidence across 186 first-party release reports and 248 post-release evaluation sources, spanning seven social-impact dimensions and multiple sectors and regions.
> >
> > Importantly, even if some trends might seem intuitive, their magnitude, consistency, and cross-provider persistence had not been documented before. Our analysis uncovers several findings that are novel, and in some cases genuinely surprising, including:
> > - Sensitive-content evaluations dominate first-party reporting, far more than other dimensions.
> > - Bias evaluations dominate third-party reporting, revealing effect sizes and patterns that were previously undocumented.
> > - Environmental-impact reporting declines sharply after late 2023, contrary to expectations given the increased public attention to compute and energy use.
> > - Some developers regress in transparency, reversing earlier good practices in subsequent model releases.
> > - A pronounced “popularity bias”, where third-party evaluations disproportionately target a small subset of widely discussed models, leaving entire classes of models unexamined.
> > - A regression of transparency. As models become more capable and widely used, first-party reporting on key social-impact dimensions, especially environmental impacts and bias, declines over time.
> > - The open paradox, where open-weight models do not exhibit stronger social-impact reporting; in fact, many have weaker documentation than their closed-weight counterparts.
> >
> > We hope that this synthesis of our findings shows that the dataset contains novel findings: several of the results above are both non-obvious and policy-relevant, and we have updated the framing in the manuscript to foreground them more clearly.
> >
> > **4. Policy Implications**
> >
> > We thank the reviewer for raising this important point about the policy implications of our work. We interpreted the reviewer’s comment as addressing two related needs:
> > (a) adding concrete policy recommendations that follow from our empirical findings, and
> > (b) providing deeper analysis of the policy drivers that shape first-party disclosure behavior.
> >
> > In response to (b), and guided by feedback from multiple reviewers, we have increased the number of interviewees and expanded our qualitative analysis of policy pressures and constraints. In response to (a), we added a new subsection on concrete policy recommendations. The revised manuscript now includes:
> > - A more detailed analysis of policy drivers identified in our interviews, including regulation-driven disclosure, organizational risk-management incentives, and competitive considerations that influence whether and how developers report social impact evaluations.
> > - A dedicated section on policy recommendations, informed by both our empirical results and interview insights. These now explicitly discuss:
> > 1) Safe-harbor policies that reduce the legal and reputational risk associated with disclosing sensitive information (e.g., emissions, data provenance).
> > 2) Standardized reporting templates aligned with ongoing governance efforts to increase comparability and reduce reporting burden.
> > 3) Publicly funded infrastructure for secure, privacy-preserving reporting, such as compute- and energy-use reporting APIs.
> >
> > We thank you again for encouraging us to make these policy implications more explicit; the revised version now better connects our empirical findings with actionable pathways for improving evaluation transparency.
> >
> > We also want to reiterate our appreciation for your detailed comments and suggestions – they helped us to significantly improve our work! Thank you!

---

### Official Review · Reviewer_8YkR · 2025-11-01

**Soundness:** 3
**Presentation:** 3
**Contribution:** 3
**Rating:** 6
**Confidence:** 3

**Summary:**

This paper presents a systematic empirical study in the field of AI governance and evaluation, analyzing how foundation models are currently assessed and reported with respect to their social impacts. Building on the seven-dimensional taxonomy proposed by Solaiman et al. (2023), the authors adopt a standardized empirical methodology that combines rigorous statistical modeling (Bayesian hierarchical ordinal regression) with cross-source data comparison. The study examines 99 first-party release-time reports and 187 third-party post-release sources to measure the prevalence and strength of social impact evaluations. Developer interviews further contextualize organizational incentives, constraints, and unobservable factors shaping reporting practices, thereby illuminating the gaps between first-party and third-party evaluations.

Results show that current evaluation practices leave major gaps in assessing the societal risks of foundation models, particularly in critical dimensions such as privacy, environmental costs, and data or content moderation labor, highlighting the need for more systematic, transparent, and comparable frameworks.

**Strengths:**

1. The paper focuses on the social impact evaluation of foundation models, an area of increasing importance within AI governance and Responsible AI research, yet still lacking systematic, quantitative evidence. The authors provide a clear empirical answer to the question "Who measures AI's social impact?" Current practices rely heavily on a small number of providers' voluntary disclosures and on third-party studies with limited access and scope. This makes the paper both policy-relevant and academically valuable.

2. Methodology combining quantitative and qualitative analyses.
    -  Quantitative component: the authors uniformly code and score 132 first-party reports and 187 third-party studies, enabling cross-provider comparability.
    -  Qualitative component: semi-structured developer interviews (from both for-profit and non-profit organizations) provide valuable contextualization, revealing the organizational incentives and constraints that shape current social impact evaluation practices.

3. It further disaggregates reporting strength across organization types (industry, academia, government, non-profit) and geographic regions (North America, Europe, East Asia, MENA, etc.), offering a fine-grained picture of provider-level variation.

4. The results are communicated with clarity:
  - First-party reporting quality is generally low and declining over time.
  - Sensitive content is the only dimension consistently evaluated, while privacy, environmental costs, and content moderation labor are largely neglected.
  - Interviews uncover the underlying incentive mechanisms, e.g., evaluations are often conducted "only when they support product adoption or compliance".
The authors propose constructive reforms such as standardized templates, incentive alignment, safe-harbor policies, and multi-stakeholder coordination.

5. The manuscript is easy to follow, and the figures and tables effectively convey the empirical trends.

**Weaknesses:**

1. The developer interview component includes only 5 participants, primarily based in the United States and France, which restricts the representativeness of the qualitative insights.

2. The paper evaluates whether social impact dimensions are reported and how detailed they are (0 - 3 scale), but does not assess whether these evaluations are scientifically valid, methodologically sound, or ethically adequate.
As a result, it primarily reveals an information disclosure gap, rather than an evaluation capability gap.

3. The study is positioned as an empirical analysis and does not propose a new theoretical framework or evaluative paradigm.

4. There appears to be a numerical inconsistency between the counts of first- and third-party reports: 99/187 in the abstract versus 132/174 in the main text.
The authors should clarify whether these refer to different units (e.g., models vs. reports) or to differences in data filtering or deduplication.

**Questions:**

See Weaknesses Section

---

> ### Author Response · Authors · 2025-12-03
> **Thank you!**
>
> We greatly appreciate the reviewer’s recognition that our work provides “a clear empirical answer to the question ‘Who measures AI’s social impact?’” and that it addresses an area of “increasing importance within AI governance.” We are grateful for the reviewer’s positive remarks regarding our combined quantitative and qualitative methodology, the clarity of our empirical findings, and the usefulness of our cross-provider comparisons. We also thank the reviewer for highlighting that the manuscript is “easy to follow”, offers a “ fine-grained picture of provider-level variation” and that the paper is both “policy-relevant and academically valuable.” We appreciate these encouraging comments and the time you invested in providing constructive feedback.
>
> **1. Limited number and geographic concentration of interviews**
>
> Following submission, we conducted additional interviews, increasing our total to 10 participants. While participants remain primarily based in North America and Europe, the expanded sample includes a more diverse set of organizational types and company sizes. We continue to make proactive efforts to recruit participants from additional geographic regions until the camera-ready version of the paper is due; however, as we now clarify in the manuscript, response rates from these regions have been substantially lower and the vast majority of currently deployed foundation models are developed in North America and Europe. This makes it challenging to obtain a regionally diverse sample. In addition, we have encountered interview restrictions from employers and identifying interview candidates that fulfilled our criteria. We are grateful to the reviewer for prompting us to articulate these limitations more clearly.
>
> **2. The paper does not evaluate the scientific validity or adequacy of the reported evaluations**
>
> We appreciate the reviewer highlighting this important distinction. We fully agree that the validity, robustness, and ethical adequacy of evaluation practices are crucial issues. At the same time, we believe that baseline transparency is the essential first step toward improving the evaluation ecosystem. Without sufficient information disclosure, assessing the quality or methodological soundness of an evaluation becomes nearly impossible. Our aim in this paper is therefore to first address the foundational problem of insufficient disclosure, which currently prevents systematic assessment of evaluation quality. We have revised the manuscript to articulate this rationale more clearly, and we thank the reviewer for encouraging us to better clarify the scope and intended contribution of our work.
>
> **3. Lack of a new theoretical framework**
>
> We thank the reviewer for raising this conceptual point. We intentionally chose not to introduce a new taxonomy or evaluative paradigm, as doing so could further fragment an already crowded conceptual landscape in proposed taxonomies. Instead, we build on and operationalize the existing framework of Solaiman et al. (2023) to produce comparable, empirical evidence. We appreciate the reviewer’s suggestion, which prompted us to elaborate on this design choice more clearly in the manuscript.
>
> **4. Numerical inconsistency between report counts**
>
> We thank the reviewer for catching this inconsistency. We have corrected and harmonized the numbers throughout the paper. The revised manuscript now clearly states that our final dataset includes:
> - 186 first-party release-time reports, and
> - 248 post-release evaluation sources, out of which 211 are fully third-party, 17 are fully first-party, and 20 are sources by model providers that report both results for their own model (labeled as first-party) and those of other providers’ (labeled as third-party).
> These numbers differ slightly from the initial write-up due to additional reports we included in response to the other reviewers’ feedback.
>
> We also want to reiterate our appreciation for your detailed comments and suggestions – they helped us to significantly improve our work! Thank you!

---

### Author Response · Authors · 2025-12-03
**Thank You & Summary of Rebuttal Process**

Dear new AC,

We sincerely appreciate the unprecedented situation you are stepping into this year. We recognize that you are volunteering your time and that the OpenReview incident has created a substantial additional burden for ACs.

We also extend our gratitude to the original reviewers and the previous AC for their thoughtful engagement and constructive feedback to our work.

Below we summarize the reviewers’ key comments and describe how we have addressed each point in our revised manuscript, which we uploaded as pdf.

**Summary of Engagement with Reviewer 8YkR**

Reviewer 8YkR emphasized several strengths of our work, noting that the paper fills a “critical gap in the current evaluation ecosystem”, which the reviewer notes is “still lacking systematic, quantitative evidence.” They found the paper to be both “policy-relevant and academically valuable” and highlighted the clarity of our mixed-methods design and presentation. Reviewer 8YkR’s concerns centered on the limited representativeness of the interview sample, the paper’s focus on disclosure rather than the validity of evaluations, and inconsistencies in reported dataset counts.

_Revisions we made in response:_
- We doubled the number of developer interviews, expanding our sample to 10 participants across non-profit, for-profit, civil-society, and academic organizations.
- We clarified the design choices and motivations underlying our methodological approach to better communicate our intent. In particular, we emphasized that our aim is to surface disclosure gaps as a necessary foundation for evaluating the validity of social-impact assessments in future work.
- We corrected a writing inconsistency the reviewer identified and harmonized numerical counts across sections.

**Summary of Engagement with Reviewer yzxU**

Reviewer yzxU recognized that our paper addresses an “important topic”, and affirmed that our annotated dataset is a “valuable contribution” to the study of social-impact evaluation practices. They encouraged clearer presentation of findings, stronger visualizations, and more explicit policy recommendations.

_Revisions made in response:_
- We updated all figures following the reviewer’s suggestions: replacing red with green where appropriate, increasing legibility, and improving temporal visualizations.
- We added summary statistics to make the quantitative findings more accessible, including distributional statistics, missingness rates, and sector-/region-level comparisons.
- We refined the exposition to highlight nuanced and previously undocumented patterns that may not be immediately intuitive, addressing the reviewer’s concern that some results seemed unsurprising.
- We added policy recommendations, including safe-harbor provisions, standardized templates, and mechanisms for cross-provider coordination, grounded in both quantitative results and interview insights.

**Summary of Engagement with Reviewer q1un**

Reviewer q1un described our work as “timely and policy-relevant”, commended our “methodological care”, and emphasized the strength of our mixed-methods triangulation and empirical evidence. They raised helpful questions regarding annotation reliability, count inconsistencies, dataset coverage, and the sensitivity of our analyses.

_Revisions made in response:_
- We added inter-annotator reliability metrics for 10% of the dataset.
- We reconciled and clarified all numerical counts.
- We expanded the dataset to include pre-2024 third-party evaluations (2018–2023), which resulted in an additional 64 annotated source documents and produced an even more pronounced temporal trend. We additionally added a flow digram as per the reviewer’s suggestion to visualize the inclusion process.
- We conducted additional sensitivity analyses, including varying hierarchical prior scales, adding provider-level slopes, and evaluating model-family effects. All core findings remained robust under these variations.

**_To the best of our knowledge, we have addressed all reviewers’ concerns and questions in our revisions, and we would have expected this to be reflected in updated assessments (original pre-rebuttal scores: 6/6/4)._**

We want to thank you all again for your engagement and your commitment to ensuring the integrity and fairness of the review process.

The authors

---

### Note · Program_Chairs · 2026-01-06
**Submission Desk Rejected by Program Chairs**

This submission has manipulated the template for wider margins and must be desk rejected.